# Reduced RNA turnover as a driver of cellular senescence

Nowsheen Mullani[1,2,*], Yevheniia Porozhan[1,*], Adèle Mangelinck[6], Christophe Rachez[1], Mickael Costallat[1], Eric Batsché[1], Michele Goodhardt[3], Giovanni Cenci[4,5], Carl Mann[6], Christian Muchardt[1]

**Accumulation of senescent cells is an important contributor to chronic inflammation upon aging. The inflammatory phenotype of senescent cells was previously shown to be driven by cytoplasmic DNA. Here, we propose that cytoplasmic double-stranded RNA has a similar effect. We find that several cell types driven into senescence by different routes share an accumulation of long promoter RNAs and 3′ gene extensions rich in retrotransposon sequences. Accordingly, these cells display increased expression of genes involved in response to double stranded RNA of viral origin downstream of the interferon pathway. The RNA accumulation is associated with evidence of reduced RNA turnover, including in some cases, reduced expression of RNA exosome subunits. Reciprocally, depletion of RNA exosome subunit EXOSC3 accelerated expression of multiple senescence markers. A senescence-like RNA accumulation was also observed in cells exposed to oxidative stress, an important trigger of cellular senescence. Altogether, we propose that in a subset of senescent cells, repeat-containing transcripts stabilized by oxidative stress or reduced RNA exosome activity participate in driving and maintaining the permanent inflammatory state characterizing cellular senescence.**

## Introduction

Cellular senescence is a state of irreversible cell cycle arrest (Rodier & Campisi, 2011). Experimentally, it can be triggered via multiple routes including prolonged maintenance in tissue culture, exposure to ionizing radiations or oxidative stress, and forced expression of mitogenic oncogenes. These inducers of senescence all share the ability to cause DNA damage and to generate reactive oxygen species, two components that probably are at the basis of the phenomenon (Ben-Porath & Weinberg, 2005). One of the

hallmarks of senescent cells is their production of a range of chemokines, pro-inflammatory cytokines, growth factors, and matrix-remodeling enzymes, defining the senescence-associated secretory phenotype or SASP. This pro-inflammatory characteristic has a crucial role in propagating senescence and in recruiting immune cells to the senescent tissue. As senescent cells accumulate with time, the SASP is also believed to be a major determinant of the chronic low-grade inflammation associated with aging and age-related diseases.

SASP activation is largely orchestrated by NF-kB and CCAAT/enhancer-binding protein beta (C/EBPb). Upstream of these transcription factors, DNA damage and the DNA damage response are major triggers of the pro-inflammatory pathways (Salminen et al, 2012). Yet, it seems that other mechanisms may allow nucleic acids to drive the chronic sterile inflammation characteristic of cellular senescence. Indeed, several studies have associated senescence with an accumulation of DNA in the cytoplasm. This DNA, in the form of chromatin, triggers innate immunity via the cytosolic DNA-sensing cGAS–STING pathway (Dou et al, 2017). Cytoplasmic DNA possibly originates from chromosome segregation errors during mitosis and its accumulation seems favored by down-regulation in senescent cells of the cytoplasmic DNases TREX1 and DNASE2 (Takahashi et al, 2018). In addition, it has been shown that in senescent cells, de-repression of repeat elements of the LINE family results in the production of retroviral RNAs, which after retrotranscription, accumulate in the cytoplasm in the form of cDNAs (Cecco et al, 2019).

Consistent with a role of cytoplasmic DNA, multiple studies have documented the importance of the interferon pathway in driving senescence, and suppression of type 1 interferon signaling hinders the onset of senescence (Katlinskaya et al, 2016). The interferon pathway was initially described as an antiviral defense mechanism activated by specific cytoplasmic or endosomal receptors of either viral DNA or dsRNA (Colby & Morgan, 1971). In this context, RNA could also be a trigger of senescence.

[1]Institut Pasteur, Centre National de la Recherche Scientifique (CNRS) UMR3738, Dpt Biologie du Développement et Cellules Souches, Unité de Régulation Epigénétique, Paris, France   [2]Sorbonne Université, Ecole Doctorale "Complexité du Vivant" (ED515), Paris, France   [3]Institut National de la Santé et de la Recherche Médicale (INSERM) U976, Institut de Recherche Saint Louis, Université de Paris, Paris, France   [4]Dipartimento Biologia e Biotecnologie "C. Darwin," SAPIENZA Università di Roma, Rome, Italy   [5]Istituto Pasteur Italia–Fondazione Cenci Bolognetti, Rome, Italy   [6]Université Paris-Saclay, Commissariat à l'Énergie Atomique et aux Énergies Alternatives (CEA), Centre National de la Recherche Scientifique (CNRS), Institute for Integrative Biology of the Cell (I2BC), Gif-sur-Yvette, France

Correspondence: christian.muchardt@sorbonne-universite.fr
Christophe Rachez, Eric Batsché, and Christian Muchardt's present address is Institut de Biologie Paris-Seine (IBPS), Institut National de la Recherche Scientifique (CNRS) UMR8256, Paris, France
*Nowsheen Mullani and Yevheniia Porozhan contributed equally to this work

To date, the role of RNA in senescence has mostly been examined at the level of discrete long non-coding RNAs (lncRNAs) regulating expression or activity of proteins relevant for cellular senescence (Montes & Lund, 2016). For example, VAD, a "very lncRNA," partially antisense to the *DDAH1* gene, inhibits the incorporation of the repressive histone variant H2A.Z at the promoter of the *INK4* gene in senescent cells (Lazorthes et al, 2015). Inversely, SENEBLOC, a Myc-regulated lncRNA, interferes with senescence in part by regulating p53-mediated repression of the *p21* gene (Xu et al, 2020). The lncRNAs HOTAIR and MALAT were also reported as regulated during entry into senescence (Tripathi et al, 2013; Yoon et al, 2013). However, some observations are suggestive of an impact of RNAs in cellular senescence at other levels, via their production, their maturation, or their turnover. For example, it was reported that neurons in the aging mouse brain accumulate 3′ UTRs, resulting in the production of small peptides of yet unknown function (Sudmant et al, 2018). Furthermore, transcripts from SINEs/Alus, a family of repetitive elements particularly abundant in euchromatin, were reported to accumulate in senescent cells with an impact on genome integrity (Wang et al, 2014). Finally, several studies have reported that cellular senescence is associated with numerous changes in the outcome of alternative pre-mRNA splicing (Wang et al, 2018). This results in the production of senescent toxins, including progerin, a variant of Lamin A associated with the Hutchinson–Gilford progeria syndrome (Cao et al, 2011), while also favoring the synthesis of S-Endoglin and p53β that may have similar pro-senescence activities (Fujita et al, 2009; Miao et al, 2016).

Here, to focus on a possible role for RNA in the triggering of the interferon pathway during the onset of senescence, we examined different senescent cells for sources of RNA species liable to activate innate immunity. This approach revealed that in several cell lines, senescence is associated with a gradual accumulation of reads originating from regions located upstream or downstream of genes and extending outside of the transposon-free regions characterizing transcription start and termination sites. Accumulation of these RNAs was correlated with reduced RNA turnover, accumulation of dsRNA in the cytoplasm, and increased activity of antiviral pathways dedicated to the degradation of dsRNAs. RNA turnover was not affected in growth-arrested cells. Consistent with a pro-senescence activity of the accumulating RNAs, inactivation of RNA exosome activity caused premature expression of senescence markers in human fibroblasts challenged with oncogenic RAF. Finally, we noted that oxidative stress caused accumulation of RNA species largely resembling those observed in senescent cells, suggesting that these RNAs participate in linking mitochondrial suffering to the permanent inflammatory state associated with senescence.

# Results

### Accumulation of RNA exosome substrates in senescent Wi38 cells

It was previously reported that in Wi38 cells driven into senescence by tamoxifen-inducible oncogenic RAF, many convergent genes displayed transcriptional termination read-through resulting in

transcriptional interference (Muniz et al, 2017). In the RNA-seq data from this study, the read-through translated into an accumulation of sequencing reads downstream of several convergent genes in the senescent cells. Re-examination of these data revealed that downstream reads accumulated at a wide range of genes beyond those described in the initial study. For example, we observed accumulation of downstream reads at several canonical histone genes (e.g., Figs 1A and S1A, green arrows). As these genes are expressed mainly in the S phase of the cell cycle and therefore strongly down-regulated in senescent cells, accumulation of these reads appeared uncoupled from the activity of the cognate gene (compare the two different scales in Figs 1A and S1A).

In the RNA-seq data, we also noted accumulation of reads downstream of several non-coding RNAs (ncRNAs). In particular, we found increased levels of 3′ extensions at essentially all U small nuclear RNA (snRNA) genes expressed in the Wi38 cells (e.g., Figs 1B and S1B, and heat map in Fig 1C). Upstream antisense RNAs (uaRNAs, also known as PROMPTs), another category of ncRNAs resulting from bi-directional initiation at promoters (schematic Fig 1D) were also accumulating at numerous genes (e.g., Fig 1A and E, blue arrows). In total, quantification at the series of 5,260 promoters not overlapping with coding regions of any gene revealed an ~10% increase in uaRNA accumulation in the senescent cells (Fig 1F and G).

Together, these observations revealed that oncogene-induced senescence in Wi38 cells is associated with an accumulation of transcripts originating from regions upstream and downstream of genes. We will refer to these RNAs as perigenic RNAs (pegeRNAs).

### Modified RNA catabolism in senescent Wi38 cells

Examination of publicly available data from HeLa cells depleted of EXOSC3 (Schlackow et al, 2017) revealed that the RNA species accumulating in senescent Wi38 cells were strikingly similar to those accumulating upon inactivation of the RNA exosome (Fig S2A–C). We therefore examined RNA exosome subunit expression in the available transcriptomic data from proliferating and senescent Wi38 cells. This allowed us to identify a strong down-regulation in the senescent cells, of *DIS3L*, encoding the catalytic subunit of the cytoplasmic RNA exosome complex. This decreased expression was confirmed by quantitative RT PCR (Fig 2A). Western blotting further revealed decreased accumulation of the DIS3L protein in the senescent cells, whereas levels of the senescence marker CDKN1A/ p21 were increased (Fig 2B). We also examined the transcriptomes for signs of activation of defense mechanisms against cytoplasmic RNA. This revealed increased expression of *OASL* and *NLRP3*, both confirmed by quantitative RT PCR (Fig 2C). OASL is associated with the antiviral OAS/RNASEL pathway detecting dsRNAs (Silverman, 2007), whereas NLRP3 is the sensor component of the NLRP3 inflammosome, functioning as a dsRNA receptor (Franchi et al, 2014).

To gain further evidence for a reduced RNA turnover in the senescent Wi38 cells, we compared accumulation of short-lived and long-lived mRNAs, the assumption being that unstable RNAs would be more affected by fluctuation in RNA exosome activity than would the stable mRNAs. For this, we established a list of highly unstable (t1/2 < 2 h) or highly stable (t1/2 > 10 h) mRNAs from a

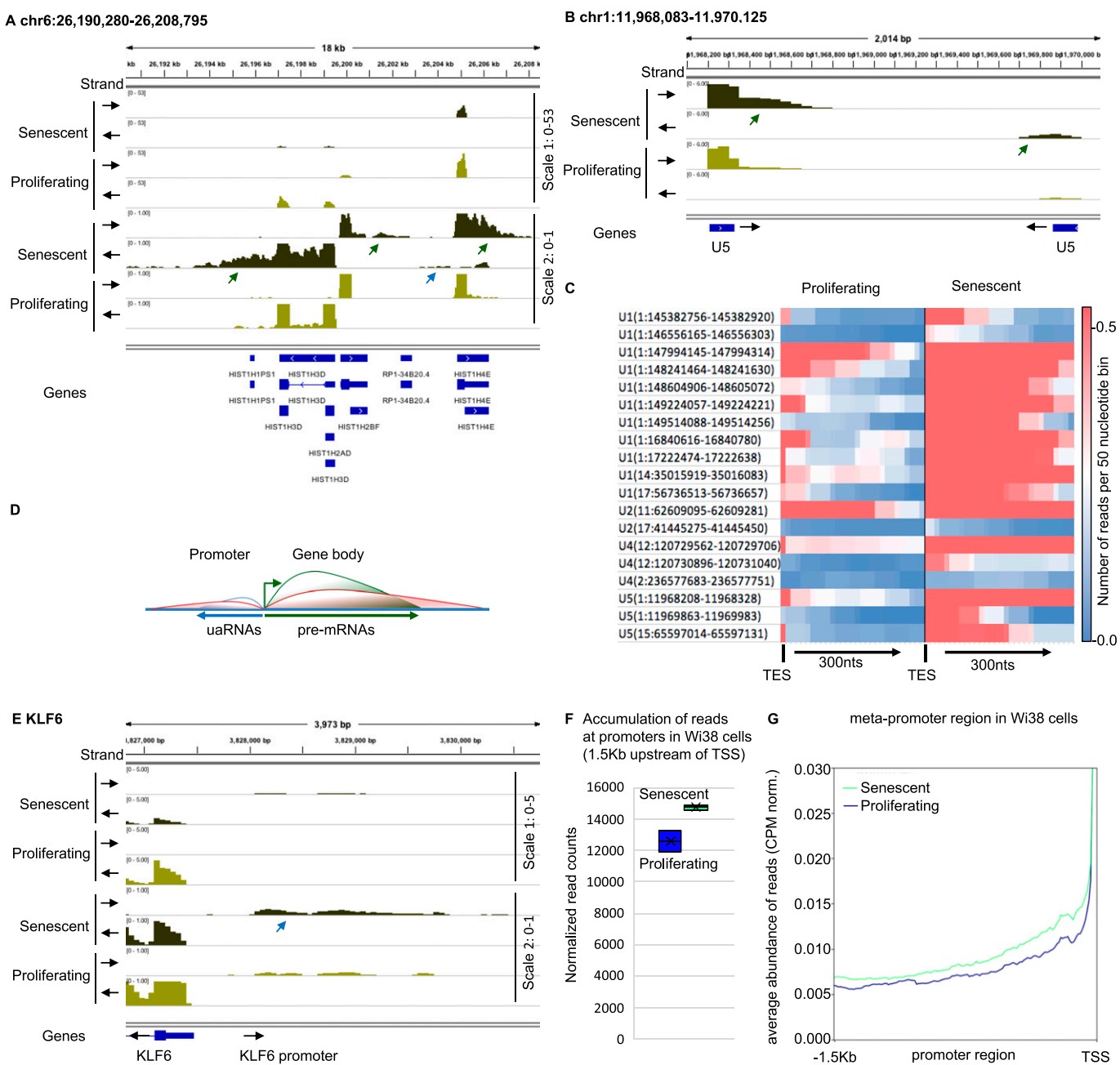

**Figure 1. Senescent WI38 cells accumulate pegeRNAs.**
RNA-seq data from WI38 hTERT RAF1-ER human fibroblasts either proliferating or driven into senescence by induction of RAF1-ER (Muniz et al, 2017). **(A, B, E)** Indicated loci were visualized with Integrative Genomics Viewer. Black arrows indicate the orientation of the track. Green arrows indicate 3′ extensions, blue arrows, promoter RNAs. **(C)** Heat map illustrating increased accumulation of 3′ extensions of U snRNAs in senescent versus proliferating WI38 hTERT RAF1-ER cells. Transcription end site indicates the 3′ end of the U snRNA gene. **(D)** Schematic representation of divergent transcription at promoters. Green line represents pre-mRNA, blue line, normal accumulation of upstream antisense RNAs, red line, accumulation of upstream antisense RNAs in senescent cells. **(F)** At 5,260 promoters not overlapping with coding regions of any gene, reads were counted within a region of 1,500 nucleotides upstream of the transcription start site (TSS) in either proliferating or senescent WI38 hTERT RAF1-ER cells. **(G)** Average profile of read distribution along the 5,260 promoters in proliferating and senescent WI38 hTERT RAF1-ER cells.

genome-wide study on mRNA stability (Tani et al, 2012). These mRNAs are listed in Table S1. We first verified our assumption by examining the effect of reduced RNA exosome activity on accumulation of these mRNAs in the HeLa cell data. This showed that

depletion of EXOSC3 resulted in increased accumulation of the short-lived RNAs when compared with WT cells. In contrast, levels of long-lived mRNAs appeared moderately decreased (Fig S2C and D). We explain this decrease by the augmented complexity of the

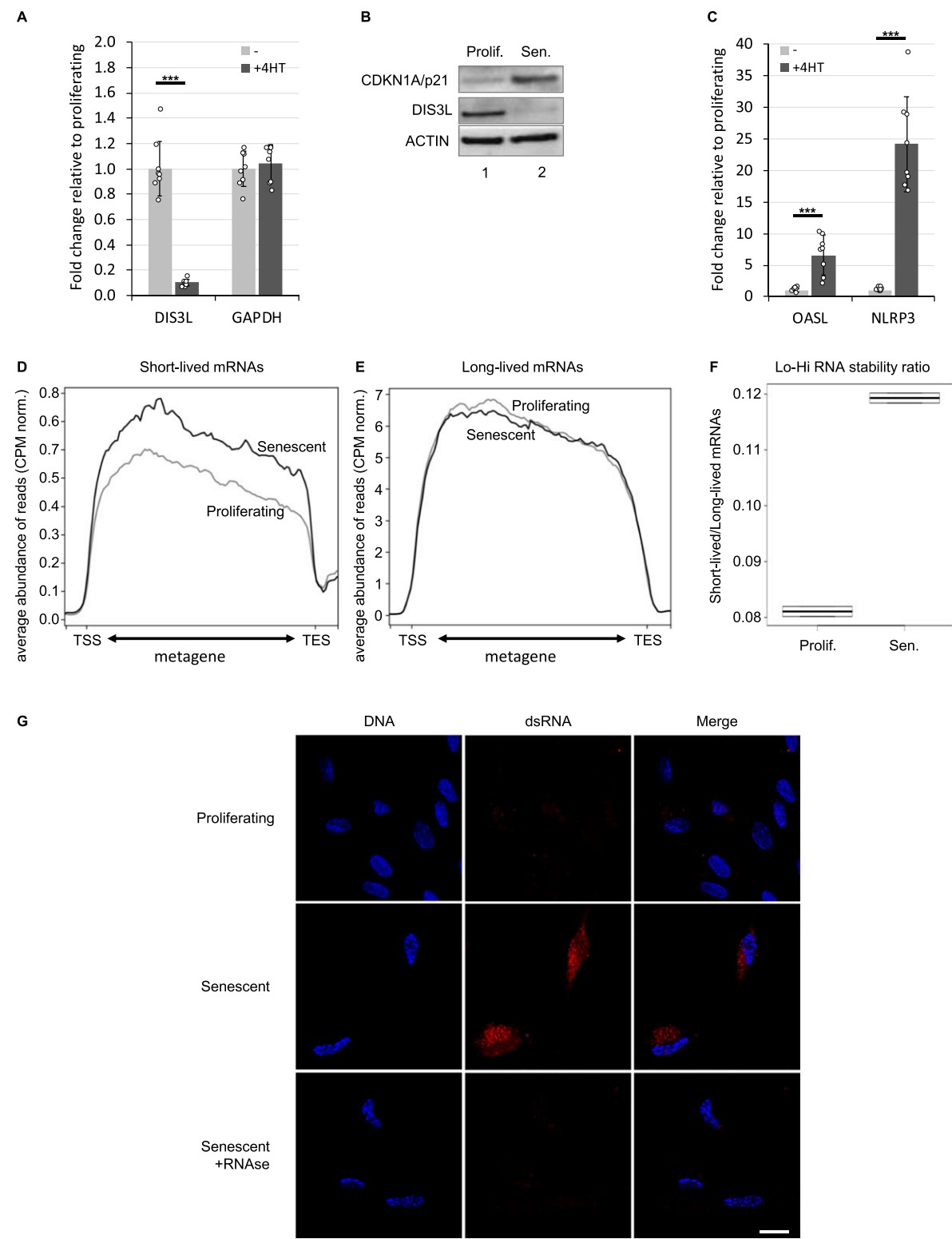

**Figure 2. Reduced expression of RNA exosome subunits in senescence.**
**(A, B, C)** Total RNA or protein extracts were prepared from WI38 hTERT RAF1-ER human fibroblasts either proliferating or driven into senescence by induction of an activated form of C-RAF. **(A, B, D)** Expression of indicated genes were assessed by quantitative RT-PCR (A, D) or Western blots (B). Indicated values were averaged from eight PCR reactions. *** indicates P-values below 0.001. **(D, E)** Average profile of reads mapping to short-lived (less than 2 h) or long-lived (more than 10 h) mRNAs as listed in Tani et al (2012), either in proliferating or in senescent cells as indicated. **(F)** Ratio of the number of reads mapping to short-lived (less than 2 h) over long-lived (more than 10 h). **(G)** WI38 hTert pTripZ bRAF^V600E cells, either proliferating or driven into senescence by inducing bRAF^V600E expression with doxycycline for 7 d, were fixed,

RNA samples in the absence of RNA exosome activity, which may result in less RNA-seq reads per RNA species. The final outcome of these variations was a 40% increase (pVal < 0.05) in the ratio of the number of reads aligning on short-lived mRNAs over those aligning on long-lived mRNAs (Fig S2E). Henceforth, this ratio will be referred to as the Lo-Hi RNA stability ratio. When applied to the Wi38 data, this approach revealed that the short-lived mRNAs acumulated more in senescent cells than in the proliferating cells, whereas levels of long-lived mRNAs remained unchanged between the two conditions (Fig 2D and E). The resulting Lo-Hi RNA stability ratio was increased by ~40%, similar to that observed in cells depleted for EXOSC3 (Figs 2F and S2E).

Finally, we investigated whether accumulating RNA species also included dsRNAs. To that end, we carried out immunofluorescent staining of Wi38 cells with the J2 anti-dsRNA IgG2a monoclonal antibody (Schonborn et al, 1991). We here used Wi38 cells expressing the activated B-RAF-V600E oncogene under the control of a tetO promoter. In our hands, this system caused a more moderate cytoplasmic contraction in association with RAF-induced senescence, as compared to the cells harboring tamoxifen-inducible RAF, and therefore allowed for easier observation of the cytoplasmic compartment. In these experiments, the J2 antibody yielded a clear RNAse-sensitive signal in the cytoplasm of the senescent Wi38 cells, suggestive of an accumulation of dsRNA in that compartment (Fig 2G).

Together, these observations argued in favor of a reduced turnover of unstable RNAs and increased accumulation of dsRNAs in the senescent Wi38 cells, in association with increased activity of innate immune defense mechanisms implicated in defense against dsRNA viruses.

## Reduced turnover of unstable RNAs in multiple senescent cells

To investigate whether the reduced turnover of unstable RNAs observed in the Wi38 cells driven into senescence by oncogenic RAF was an exception or a widespread phenomenon, we examined several additional RNA-seq data sets from senescent cells.

First, we examined another case of oncogene induced senescence involving expression of oncogenic RAS in IMR90 human fibroblasts. In this series, RNA-seq (n = 2) was carried out after expression of RAS for 0, 4, or 10 d (Lau et al, 2019). Examination of the transcriptome of the cells revealed a progressive reduction in transcripts from several genes encoding subunits of the RNA exosome, including DIS3L, EXOSC2, EXOSC3, EXOSC6, EXOSC8, EXOSC9, and EXOSC10 (Fig 3A). Three subunits were unaffected (DIS3, EXOSC1, and EXOSC7), whereas two non-catalytic subunits were up-regulated (EXOSC4 and EXOSC5). This data set was of insufficient depth to visualize pegeRNAs; we therefore examined (1) the accumulation of unstable compared with stable mRNAs, and (2) the expression of the OAS enzymes. Consistent with a reduced RNA turnover, we found that the Lo-Hi RNA stability ratio increased progressively during the entry of the IMR90 cells into senescence (Fig 3B and C). OAS1 and OAS2 mRNA levels

increased accordingly, being, respectively, up-regulated 316-fold and 74-fold at 10 d (Fig 3D). In parallel, expression of DNASE2 and of the DNA receptor TLR9 were increased twofold and sixfold, respectively, at 10 d, whereas the DNAse TREX1 was essentially unaffected. The variations of DNASE2 and TLR9 expression suggest that that these cells may accumulate simultaneously both cytoplasmic DNA and dsRNA.

Next, we examined a data set from human fibroblasts (HCA-2 cells), keratinocytes, and melanocytes driven into senescence by ionizing radiation (n = 6, with data points at 0, 4, 10, and 20 d of culture) (Hernandez-Segura et al, 2017). In this series, the Lo-Hi RNA stability ratio was gradually increased only in the melanocytes (Fig 3E). These cells were also the only ones to up-regulate the OAS genes, consistent with them having to cope with accumulation of dsRNAs (Fig 3F, 106-fold for OAS1, 182-fold for OAS2, 33-fold for OAS3, and 397-fold for OASL at 20 d). Transcription of the RNA receptors TLR3, IFIH1/MDA5, and DDX58/RIG-I was also increased at all time points in the melanocytes, whereas the HCA-2 fibroblasts up-regulated the DNA receptor TLR9 (Fig 3G). Of note, in the melanocytes, it was not possible to unambiguously associate the suspected reduced RNA turnover with reduced transcription of any specific RNA exosome gene. This will be further addressed in the discussion. Finally, this data set also included RNA-seq from quiescent fibroblasts, allowing us to verify that reduced RNA turnover was not triggered by simple growth arrest (Fig 3E–G).

To further probe the robustness of our observations, we finally examined a large data set from Wi38, IMR90, HAEC, and HUVEC cells driven into senescence via different routes (n = 2 for each condition), including oncogenic RAS, replicative exhaustion, and DNA damage by treatment with either doxorubicin or γ irradiation (Casella et al, 2019). In all experiments on Wi38 cells, senescence led to an increased Lo-Hi RNA stability ratio (Fig S3A). Consistent with this, expression of the OAS genes was up-regulated upon replicative and DNA damage-induced senescence (Fig S3B, top panel). This activation of the OAS genes correlated with increased expression of the TLR3, IFIH1/MDA5, and DDX58/RIG-I sensors of dsRNA (Fig S3B, bottom panel). Interestingly, in the RAS-induced Wi38 senescence, like in the initial RAF-induced Wi38 senescence (Fig 2C), transcriptional activation was restricted to OASL and the inflammosome component NLRP3 (Fig S3B, column 1). In all cases, the DNases TREX1 and DNASE2 were affected less than twofold although we noted a three to fourfold activation of the DNA receptor TLR9 in two of the data sets (Dox(2) and 10 Gy).

From the same data set, we next analyzed the RNA-seq from three different cells types, HAEC, HUVEC, and IMR90 cells driven into senescence by γ irradiation or replicative exhaustion. We observed that the Lo-Hi RNA stability ratio was only moderately increased for the HAEC and HUVEC cells, whereas it was decreased in the IMR90 cells (Fig S3C). The modestly increased Lo-Hi RNA stability ratio in the HAEC and HUVEC cells translated into a small but significant (pVal < 0.05) increase in OAS1, OAS2, TLR3, and IFIH1/MDA5 gene

permeabilized, and analyzed for dsRNA using the mouse monoclonal antibody J2 (red). To visualize the cell nuclei, DNA was stained with DAPI (blue). Where indicated, fixed and permeabilized cells were treated with a cocktail of RNAseA and RNAseH before incubation with the J2 antibody. Scale bar: 5 $\mu$m. Source data are available for this figure.

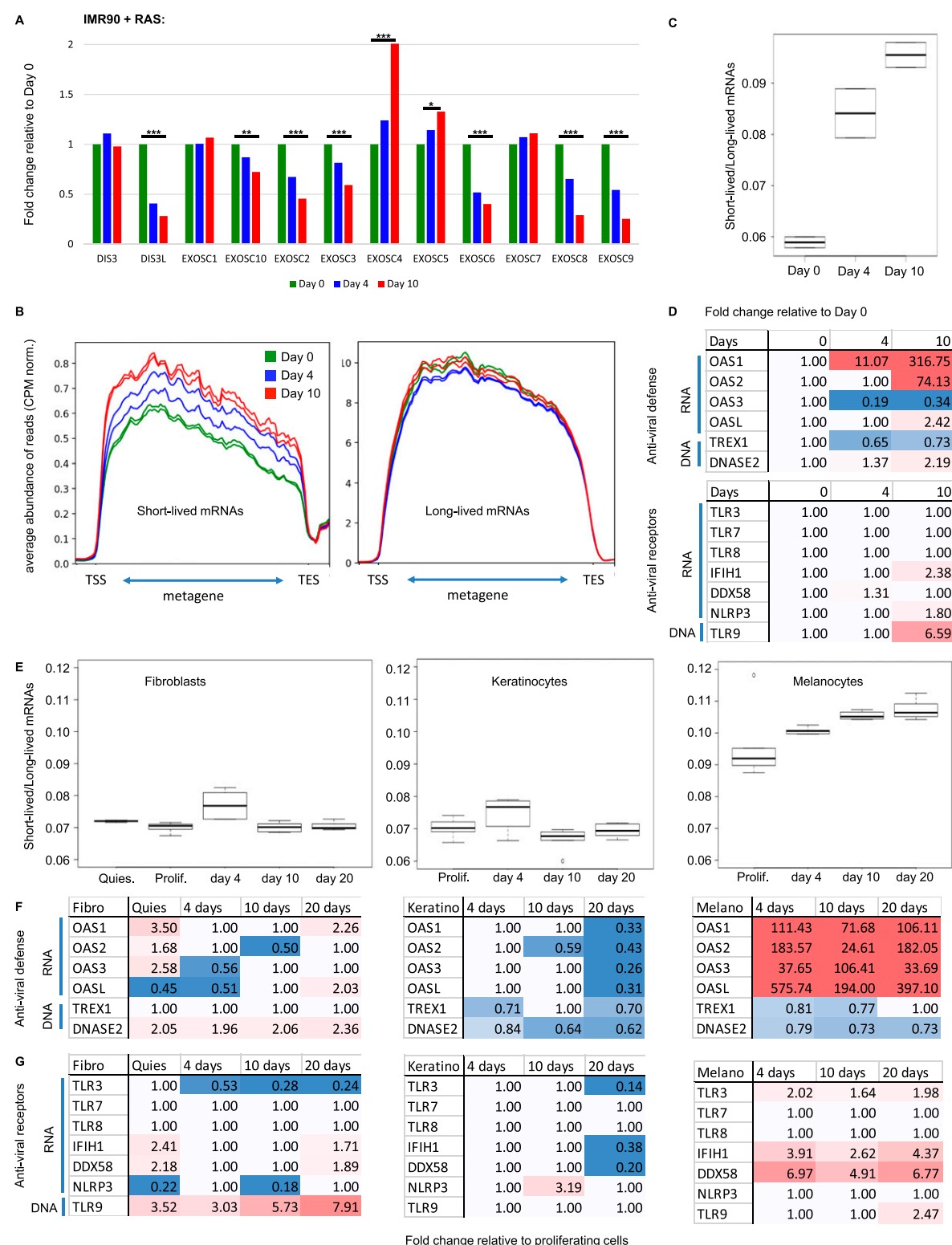

**Figure 3.   Reduced RNA decay in senescent cells of various origin.**
**(A, B, C, D)** RNA-seq data from IMR90 human fibroblasts induced into senescence via activation of the oncogene Ras (Lau et al, 2019). Samples (n = 2) had been collected at day 0 (growing phase), day 4 (beginning of SASP induction) and day 10 (senescent phase) of Ras induction. **(A)** Histograms show variations in expression levels of the indicated genes. *** and ** indicate *P*-values below 0.001 and 0.01, respectively. **(B)** Average profile of reads mapping to short-lived (less than 2 h) or long-lived (more than 10 h) mRNAs as in Fig 2 at the indicated time points. **(C)** Lo-Hi RNA stability ratio for the experiment at the indicated time points, calculated as in Fig 2. **(D)** Fold activation of the indicated genes at the indicated time point. Red gradient indicates up-regulation (max red at 10-fold), blue indicates down-regulation (max blue at 0.5). Variations

expression, whereas these genes were unaffected or down-regulated in the IMR90 cells (Fig S3D).

Altogether, these observations identify a subcategory of senescent cells displaying increased accumulation of the more unstable RNAs and signs of a defense reaction against dsRNA. The variability in transcriptional landscapes observed in the different senescent cells support a model where each cell line is affected by accumulating RNAs to different degrees, possibly as a complement to cytoplasmic accumulation of DNA.

## Senescent cells share an RNA signature with cells exposed to oxidative stress

Accumulation of uaRNAs were previously observed in cells exposed to oxidative stress, a central determinant of senescence (e.g., *KLF6* locus in Fig S4A and B and Giannakakis et al [2015], Nilson et al [2017]). This phenomenon was imputed to defective RNA polymerase II transcription termination (Nilson et al, 2017). To investigate possible similarities between oxidative stress and senescence at the level of the RNAs accumulating in the cells, we examined cells exposed to $H_2O_2$ for the presence of pegeRNAs other than uaRNAs. For this study, we chose a data set from $H_2O_2$-treated BJ or MRC5 cells allowing for detection of rare RNAs (Giannakakis et al, 2015). In these data, like in the RAF-induced Wi38 senescent cells and the EXOSC3-depleted HeLa cells, we observed an accumulation of reads downstream of many histone genes (Figs 4A and S4C). We also observed the accumulation of 3′ extensions of U snRNAs (see example in Figs 4B and S4D, and heat map of Fig 4C). Finally, as in the senescent cells, we observed a gradual increase in the Lo-Hi RNA stability ratio, suggestive of a reduced RNA turnover (Figs 4D and S4E).

We next examined the transcriptome of a mouse model involving oxidative stress associated with mitochondrial dysfunction caused by inactivation of the Mof histone acetylase in the heart (Chatterjee et al, 2016). Inactivation of this gene has severe consequences on cardiac tissue that has high-energy consumption, triggering hypertrophic cardiomyopathy and cardiac failure. At the cellular level, cardiomyocytes were reported to show severe mitochondrial degeneration and deregulation of mitochondrial nutrient metabolism and oxidative phosphorylation pathways. Consistent with mitochondrial suffering and ensuing oxidative stress, Mof inactivation reproduced the increased accumulation of uaRNAs observed in the $H_2O_2$-treated human tissue-culture cells (see example of the *Ryr2* promoter, Fig 4E, and metaplot of 1,200 promoters, Fig 4F). Likewise, *Mof* inactivation resulted in accumulation of 3′ extensions at U snRNA and histone genes, reproducing another RNA signature of senescence (Fig 4G and H). Consistent with reduced RNA decay, we noted in the *Mof* KO cells, a significant decrease in the expression of several subunits of the RNA exosome, particularly the catalytic subunit Dis3l, as well as Exosc3, Exosc7, and Exosc9 (Fig 4I). In parallel, and consistent with *Mof* inactivation causing oxidative stress and DNA damage, we

noted a clear increase in the expression of the senescence marker *Cdkn1a/p21* (Fig 4J).

Together, these observations suggest that the oxidative stress associated with cells undergoing senescence may be upstream of the accumulation of pegeRNAs observed in these cells.

## RNA exosome depletion accelerates the onset of senescence

To investigate whether inactivation of the RNA exosome had a causative impact on the onset of senescence, Wi38 cells harboring the tetO-B-RAF-V600E construct were transfected with EXOSC3 siRNAs. As previously described, the tetO-B-RAF-V600E system allows modulating the speed at which cells are induced into senescence by reducing the concentration of doxycyclin (Carvalho et al, 2019). Thus, cells received 25 ng/ml doxycycline for 0, 2, or 3 d (see schematic Fig 5A and EXOSC3 depletion Fig 5B and C). This short treatment at reduced doxycycline concentrations is not sufficient to induce senescence and did not result in induction of *CDKN1A/p21* and *CDKN2A/p16* and in only a 10-fold induction of *CDKN2B/p15* after 3 d (Fig 5D–F, light-gray bars). In contrast, in cells depleted for EXOSC3, *CDKN2A/p16*, and *CDKN2B/p15* were up-regulated 6-fold and 25-fold, respectively, by day 2, suggesting that reduced RNA turnover facilitates the onset of oncogene-induced senescence (Fig 5D–F, dark-gray bars). We note, however, that EXOSC3 depletion alone, in the absence of B-RAF-V600E expression, did not induce senescence markers within the time frame of the experiment.

To gain further evidence for an impact of RNA exosome inactivation on cellular senescence, we examined a data set from mouse embryonic stem cells depleted in Exosc3 for 3 d (Chiu et al, 2018). As expected and as described in the original article, these cells recapitulated the increased accumulation of uaRNAs (e.g., Fig 5G). They also recapitulated the effect on U snRNA maturation (e.g., Fig 5H). Gene Ontology (GO) term analysis of genes up-regulated upon Exosc3 inactivation further revealed a highly significant enrichment in genes associated with the p53 pathway and we noted increased expression of the senescence markers *Cdkn2b/p15*, *Cdkn2a/p16*, and *Cdkn1a/p21* (Fig 5I and J and Table S2 listing differentially regulated genes associated with the GO terms). Consistent with this, down-regulated genes were highly enriched in genes associated with cell cycling (Fig 5K). These observations show that loss of RNA exosome activity results in a transcriptional landscape sharing many characteristics with senescent cells.

Examining GO terms for cellular compartments further highlighted a significant enrichment in mitochondrial genes among the genes down-regulated by depletion of Exosc3 (Fig 5L). This was consistent with an earlier report showing mitochondrial dysfunction in patients with pontocerebellar hypotrophy, a syndrome linked to a mutation in the *EXOSC3* gene (Schottmann et al, 2017).

Together, these observations indicate that reduced RNA exosome activity induces transcriptional traits characteristic of cellular senescence, possibly favoring the onset of senescence. In addition, the data are suggestive of a bidirectional crosstalk between RNA degradation and oxidative stress for the induction of growth arrest.

---

with pVal > 0.05 were set to 1. **(E, F, G)** RNA-seq data from Hernandez-Segura et al (2017), n = 6 for each cell type and time point. HCA-2 (fibroblasts), keratinocytes, or melanocytes had been exposed to ionizing radiation. RNA had been harvested 4, 10, or 20 d later. **(E)** Lo-Hi RNA stability ratio for each experiment calculated as indicated in Fig 2. **(D, F, G)** Fold activation of the indicated genes represented as in (D).

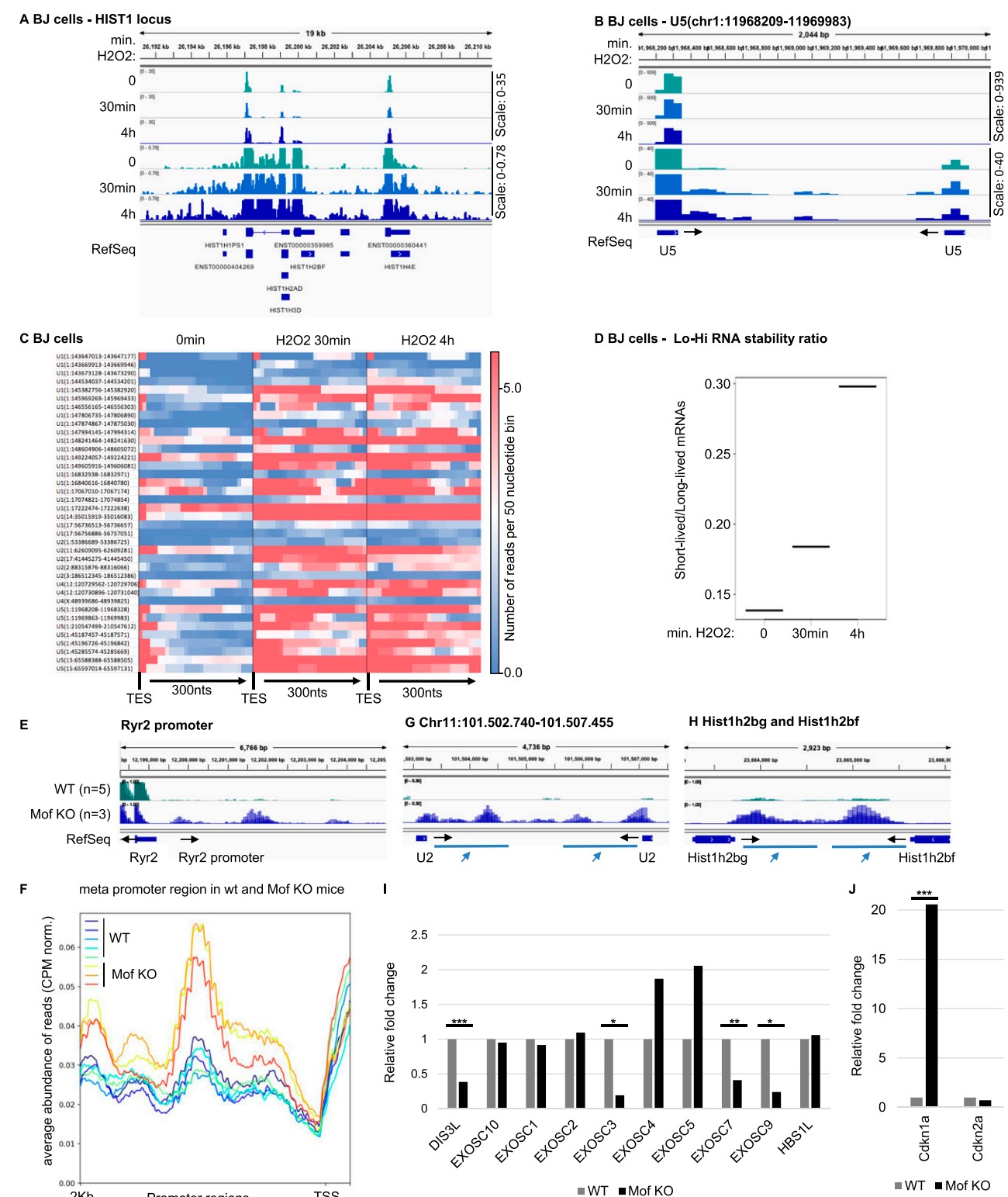

**Figure 4. Accumulation of pegeRNAs in cells exposed to oxidative stress.**
**(A, B, C, D)** BJ human fibroblasts exposed to 0.2 mM $H_2O_2$ for the indicated times (one sample per time point) (Giannakakis et al, 2015). Indicated loci were visualized in the Integrative Genomics Viewer genome browser. **(C)** Heat map illustrating increased accumulation of non-maturated U snRNAs in BJ cells exposed to $H_2O_2$ for the indicated times. Transcription end site indicates the 3′ end of the U snRNA gene. **(D)** Lo-Hi RNA stability ratio for BJ cells at the indicated time points, calculated as indicated in Fig 2. **(E, F, G, H, I, J)** Mouse cardiomyocytes either WT (n = 4) or inactivated for the Mof histone acetylase (n = 3) were analyzed by RNA-seq (Chatterjee et al, 2016). **(E, G, H)** Data were aligned on mouse genome mm9 and indicated loci were examined using the Integrative Genomics Viewer genome browser. **(F)** Average profile of read distribution along the promoters of 1,200 genes with similar expression levels in either WT or Mof KO cardiomyocytes.
**(I, J)** Differential gene expression was estimated with DESeq2. Histograms show variations of the indicated genes. ***, **, and * indicate *P*-values below 0.001, 0.01, and 0.05 respectively.

### Serendipitous transcription of DNA repeats in pegeRNAs

We finally investigated how pegeRNAs may trigger cytoplasmic antiviral mechanisms. In this context, we noted that an earlier study had reported an accumulation in senescent cells of RNAs originating from the SINE family of retrotransposons (Wang et al, 2014). In parallel, another study showed that SINE-containing RNAs can trigger an interferon response and stimulate secretion of cytokines (Hung et al, 2015). Together, these observations prompted us to investigate whether pegeRNAs may be enriched in SINE or other repeated sequences prone to yield dsRNAs detectable by the interferon pathway.

To investigate this possibility, the average distribution of SINEs and LINEs in regions producing pegeRNAs was examined by plotting the profile of these repetitive elements over all protein-coding genes (annotated NM in the RefSeq database). This revealed that the density in SINEs and LINEs was low at transcription start site (TSS) and transcription end site, but rapidly increased when moving away from these sites (Fig 6A). Likewise, density in SINEs and LINEs was low over U snRNA gene bodies, but then returned to average levels within a few hundred nucleotides from the gene boundaries (Fig 6B). Thus, SINE and LINE repeats have a distribution compatible with them being absent from short promoter RNAs and short 5′ extensions, but present within the longer pegeRNAs.

To document this, we plotted the average distribution of reads at promoters in Wi38 cells driven into senescence by oncogenic RAF (Fig 6C), or in BJ cells exposed to $H_2O_2$ (Fig 6D). In these plots, we scaled the region from the TSS to the first SINE to a fixed 1 Kb. This allowed us to visualize transcription before and after the first SINE upstream of the TSS. In both data sets, promoter transcription rarely reached beyond the first SINE in proliferating cells (Fig 6C and D, dark-blue lines), whereas this apparent first-SINE limit was readily crossed in the senescent cells and even more so in the cells exposed to oxidative stress (Fig 6C and D, light-colored lines).

These observations define pegeRNAs as a source of transcripts containing transposon sequences, and we speculate that these transposons could be a source of dsRNA detected by the antiviral defense mechanisms.

## Discussion

Senescent cells have many characteristic phenotypes, including growth arrest, modified chromatin structure, and secretion of pro-inflammatory molecules. Here, we document an additional characteristic, namely, the accumulation of RNA species transcribed from regions located upstream and downstream of genes. These RNA species that we refer to as perigenic RNAs (pegeRNAs), include uaRNAs and 3′ extensions of genes. These RNAs were found to accumulate in multiple cell lines, driven into senescence via multiple routes, including forced expression of oncogenes, DNA damage, and replicative exhaustion.

3′ extensions of genes are removed from the main transcript during U snRNA maturation. The cleaved product is then degraded by the RNA exosome (Allmang et al, 1999). Accumulation of reads downstream of genes in HeLa cells depleted in EXOSC3, as well as

observations in yeast (Lemay et al, 2014; Villa et al, 2020), suggest that the RNA exosome also participates directly or indirectly in transcription termination of mRNAs. Finally, the RNA exosome clears the cells of uaRNAs (Ogami et al, 2018). Thus, the different RNA species composing pegeRNAs are all likely targets of this RNA degradation machinery. Accordingly, we observed reduced expression of RNA exosome subunits in two of the senescent cell lines, namely, the Wi38 and the IMR90 cells driven into senescence by oncogenic RAF and RAS, respectively. In both cases, expression of the *DIS3L* gene coding for the catalytic subunit of the cytoplasmic RNA exosome was affected. Expression of this subunit was also down-regulated in the cells exposed to chronic oxidative stress in the mouse model inactivated for Mof. In the IMR90 and in the mouse cells, expression of the regulatory subunits EXOSC3 and EXOSC9 were also down-regulated. In contrast, the non-catalytic subunit EXOSC4 was up-regulated in all three cellular models, for reasons which are as yet unclear.

In the other senescent cells examined, expression of *DIS3L* was down-regulated less than twofold. We speculate that in these cells, the reduced RNA decay may also involve post-translational mechanisms. The existence of such mechanisms is suggested by the very rapid accumulation of pegeRNAs (30 min) observed in cells exposed to $H_2O_2$ (this article and original analysis by Giannakakis et al [2015]). Similarly, the half-life of unstable ncRNAs was previously shown to be drastically increased in cells exposed to oxidative stress (Tani et al, 2019).

To estimate RNA decay efficiency, we took advantage of the differential sensitivity of stable and unstable mRNAs to changes in the activity of RNA degradation machineries. This approach was verified in cells with reduced RNA exosome activity caused by depletion of EXOSC3. Calculating the ratio between reads mapping to unstable RNAs and stable RNAs (Lo-Hi RNA stability ratio) allowed us to address changes in RNA decay efficiency even in RNA-seq data sets of insufficient depth for the detection of the rare pegeRNAs or carried out on poly(A) selected libraries. This Lo-Hi RNA stability ratio does not provide any direct information on the activity of the RNA exosome, but may be of general interest as a first approach to the analysis of RNA decay in, for example, cancer cells.

An increased Lo-Hi RNA stability ratio was associated with increased expression of *OAS* genes in essentially all the senescent cells we examined. We speculate that this is a manifestation of cells facing accumulation of dsRNAs in the cytoplasm. Indeed, we observed dsRNAs accumulating in the cytoplasm of senescent Wi38 cells. In addition, an earlier study has shown that pegeRNAs are extensively detected in the polysomes of cells exposed to $H_2O_2$ (Giannakakis et al, 2015). Possibly, the high stability of these RNAs in cells exposed to stress may increase the probability of their export from the nucleus to the cytoplasm.

Examination of the transcriptomes also provided clues as to the receptors possibly involved in the detection of the stabilized RNAs. Indeed, we noted that up-regulation of the interferon-inducible *OAS* genes was frequently associated with increased expression of either *TLR3*, *IFIH1/MDA5*, and *DDX58/RIG-I* genes, which encode RNA sensors upstream of the interferon pathway, or of NLRP3, the inflammosome sensor protein. Although the regulation of these RNA sensors is poorly documented, their increased transcription may reflect an increased need for RNA detection in the lengthy process of entering senescence, or it may represent an auto-amplification mechanism.

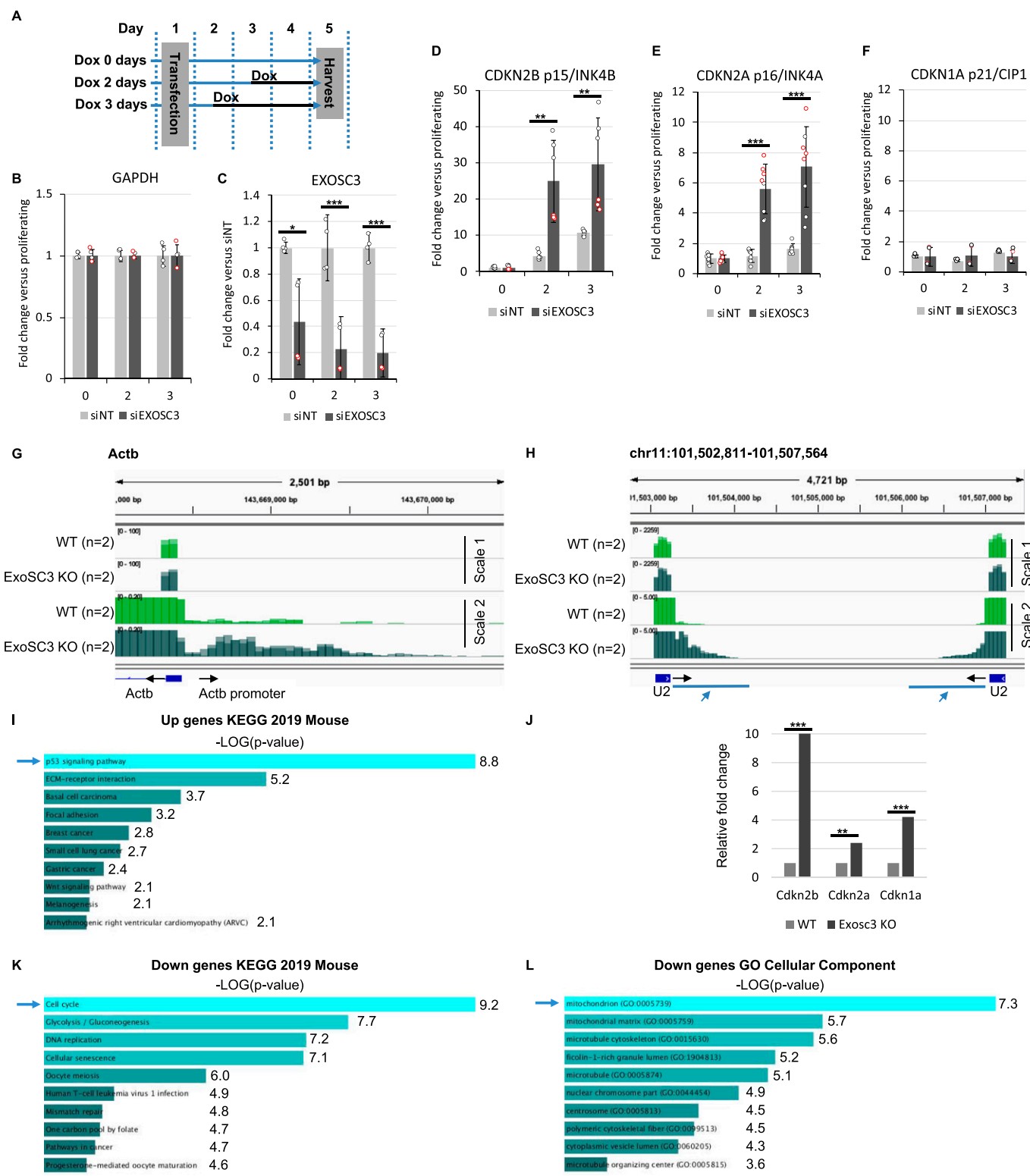

**Figure 5. Inactivation of the RNA exosome induce a senescent-like phenotype and deregulates mitochondrial genes.**
**(A)** Schematic: WI38 hTert pTripZ bRAF[V600E] cell were transfected with either non-targetted siRNAs (siNT) or two different EXOSC3 siRNAs (red and white dots in histograms), then treated with 25 ng/ml of doxycycline at the time indicated in the schematic. All samples were harvested and the same time, and total RNA was extracted. **(B, C, D, E, F)**, Expression of indicated genes were assessed by quantitative RT-PCR. Indicated values were averaged from eight PCR reactions. ***, and ** indicate *P*-values below 0.001, and 0.01 respectively. **(G, H, I, J, K, L)** RNA-seq data (n = 2) from mouse ES cells inactivated for Exosc3 and harboring an inducible Exosc3 expression construct human (Chiu et al, 2018). Exosc3 expression is initially induced (WT) then the induced is removed from the medium and cells are cultured for 3 d (Exosc3 KO). **(G, H)** Indicated loci were visualized with Integrative Genomics Viewer. Black arrows indicate the orientation of the gene. Blue arrows indicate regions of interest. **(I, K, L)** GO term analysis of genes differentially expressed upon Exosc3 KO was carried out with Enrichr. **(J)** Differential gene expression was estimated with DESeq2. Histograms show variations of the indicated genes. ***, and ** indicate *P*-values below 0.001, and 0.01 respectively.

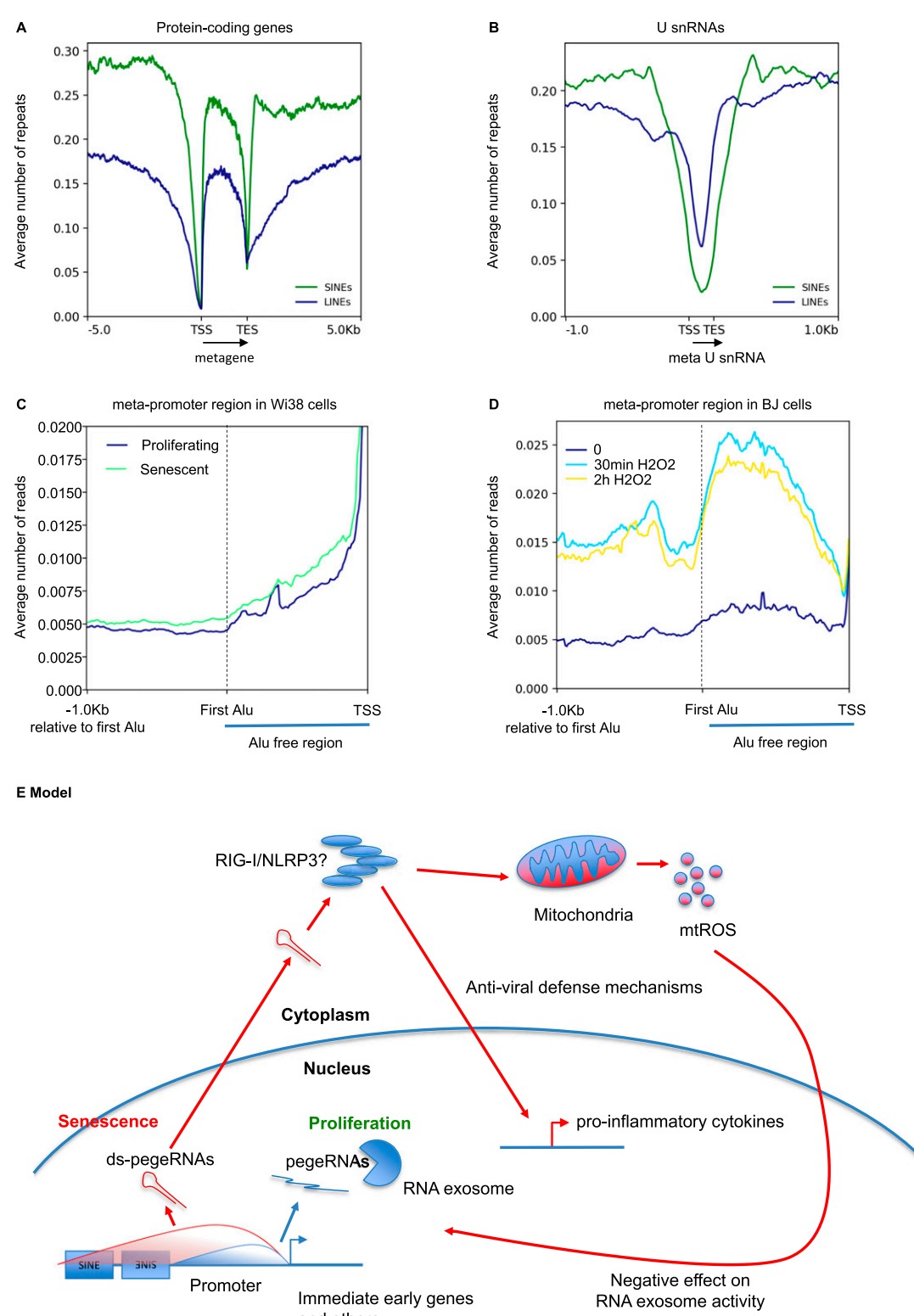

**Figure 6.  pegeRNAs from senescent cells reach into SINE- and LINE-containing regions.**
**(A, B)** Average number of either SINEs (green profiles) or LINEs (blue profiles) per 50 nucleotide bin in the neighborhood of either protein-coding genes (A) or snRNAs (B). TSS, transcription start site; TES, transcription end site. Gene bodies are all scaled to 2 Kb. U snRNA bodies are scaled to 200 nucleotides. Location of SINEs and LINEs was obtained from the Hg19 version of RepeatMasker. **(C, D)** At the 5,260 promoters not overlapping with coding regions of any gene described in Fig 1, the region from the TSS to the first Alu sequence was scaled to 1 Kb (indicated as Alu-free region). The profile then continues 1 Kb after the start of this first Alu. The average number of reads per 50 nucleotide bin was then calculated within and upstream of this region. **(C, D)** This was carried out for the RNA-seq data from (C) WI38 hTERT RAF1-ER human fibroblasts

NLRP3 was reported to be activated by Alu RNAs (Tarallo et al, 2012). RNAs encoded by these retrotransposons have previously been shown to stimulate secretion of cytokines (Hung et al, 2015). This emphasizes the possible role of repeated motifs originating from retrotransposons in the activity of pegeRNAs. In senescent cells and cells exposed to oxidative stress, pegeRNAs will harbor sequences from transposable elements as the transcripts extend beyond the repeat-free promoter and termination regions. In turn, these sequences will potentially be subject to direct detection by, for example, Ro60/TROVE2 that binds inverted Alu sequences (Hung et al, 2015), or will fold into dsRNAs if the transposons form inverted repeats.

Among the senescent cells we examined, we also identified four cases for which the Lo-Hi RNA stability ratio was either unchanged or decreased, namely the fibroblasts IMR90 and HCA-2, and the keratinocytes driven into senescence by γ irradiation, and the IMR90 fibroblasts subject to replicative senescence. In these cells, the DNA/cGAS/STING1 pathway may be dominant in triggering the inflammatory phenotype. This leads us to propose that DNA and pegeRNAs may be two alternative and possibly complementary drivers of senescence.

An active role for pegeRNAs in the onset of senescence is suggested by the accelerated production of senescence markers that we observed in Wi38 cells depleted in EXOSC3 with siRNAs and then challenged with activated RAF. In this experiment, we also noted that EXOSC3-depletion alone was not sufficient to trigger the senescence markers. This may, however, be due to the transient nature of siRNA depletion. Indeed, mining of publicly available data showed that inactivation of the *Exosc3* gene in mouse ES cells triggered several senescence associated phenotypes, including activation of *Cdkn1a/p21*, *Cdkn2a/p16*, and *Cdkn2b/p15*, activation of the p53 pathway, and reduced expression of multiple markers of cell cycle progression, all in favor of a triggering effect of pegeRNA.

Inactivation of Exosc3 in the mouse ES cells resulted also in reduced expression of mitochondrial genes. Interestingly, medical data recapitulate this observation. Indeed, patients suffering from pontocerebellar hypoplasia, a disease affecting the development of the brain and due to a mutation in the *EXOSC3* gene were also described as showing signs of mitochondrial dysfunction (Schottmann et al, 2017). In parallel, patients with mutations in *EXOSC2* are subject to premature aging (Di Donato et al, 2016). TLR3, IFIH1/MDA5, DDX58/RIG-I, and NLRP3 are all connected with mitochondria (Djafarzadeh et al, 2011; Zhou et al, 2011), whereas mitochondrial dysfunction is a source of oxidative stress, which in turn is a trigger of pegeRNA accumulation. We suggest that together, these elements compose a possible feedback loop in which accumulation of pegeRNAs triggers antiviral defense mechanisms causing production of mitochondrial reactive oxygen species that in turn nurtures both the inflammatory response

and the reduced RNA turnover. Once initiated, such a process could be an irreversible driver of cellular senescence (Model Fig 6E).

# Materials and Methods

### Tissue culture

WI38 hTERT RAF-ER cells, which are immortalized by hTERT expression and contain an inducible *RAF1* oncogene fused to the estrogen receptor (ER), were maintained in MEM supplemented with glutamine, nonessential amino acids, sodium pyruvate, penicillin–streptomycin, and 10% fetal bovine serum in normoxic culture conditions (5% $O_2$) (Jeanblanc et al, 2012). For the induction of oncogene-induced senescence, the cells were treated with 20 nM 4-HT (H7904; Sigma-Aldrich) for 3 d. WI38 hTert pTripZ bRAFV600E cells that express bRAF$^{V600E}$ under the control of a tetO promoter were cultivated likewise. RAF expression was induced by either 25 ng/ml (siRNA experiment) or 200 ng/ml (immunofluorescence) of doxycycline (D3447; Sigma-Aldrich).

### Immunofluorescent staining

Proliferating WI38 hTert pTripZ bRAFV600E cells, and the same cells induced into senescence by incubation with 200 ng/ml of doxycycline for 7 d, were washed with PBS and then fixed with 1.6% formaldehyde for 15 min. Cells were grown on collagen-coated cover slips in 24-well plates. Cells were washed twice with PBS and permeabilized by incubation with PBS 0.2% Triton X-100 for 5 min. One set of senescent cells was then treated with a mix of 40 U/ml RNaseA + 0.25 mg/ml RNaseH in PBS + 50 mM MgCl2 for 3 h at 37°C. All samples were then washed three times with PBS + 0.1% Tween-20 + 0.05% Triton X-100 and then incubated with a blocking buffer consisting of PBS + 5% BSA + 0.1% Tween-20 for 1 h at RT. Cells were then incubated with a 1/200 solution of mouse monoclonal J2 anti-dsRNA antibody (Scicons) for 1 h 30 min at RT. in blocking buffer. Cells were washed three times with PBS + 0.1% Tween-20 + 0.05% Triton X-100 and then incubated with a 1/500 dilution of Alexa-594 antimouse secondary antibody in blocking buffer for 1 h at RT. Cells were washed three times with PBS + 0.1% Tween-20 and then stained for 5 min with 0.25 µg/ml DAPI in PBS. Cells were washed once with PBS and then mounted in an antifading mounting solution on microscope slides. Images were acquired with a Leica SP8 confocal microscope with a 63× objective. The images were all acquired with identical settings.

---

either proliferating or driven into senescence (Lazorthes et al, 2015) or (D) BJ cells exposed to $H_2O_2$ for the indicated times (Giannakakis et al, 2015). **(E)** Hypothetical model: an initial source of oxidative stress increases elongation and reduces degradation of pegeRNAs that will eventually contain sequences encoded by repeats originating from retrotransposons. Because of their abundance, a fraction of the pegeRNAs reaches the cytoplasm (Giannakakis et al, 2015). In the cytoplasm, dsRNAs generated by inverted repeats are detected by antiviral defense mechanisms. Activation of these RNA receptors results in mitochondrial dysfunction (Djafarzadeh et al, 2011). This leads to production of mitochondrial reactive oxygen species (mtROS) that hampers the RNA exosome activity and feeds the inflammatory phenotype of senescent cells.

## Western blotting

Cells were lysed with lysis buffer (50 mM Tris–HCl, pH 7.5, 150 mM NaCl, 1% Triton X-100, 0.1% SDS, 1 mM EDTA, and protease/phosphatase inhibitor mixture [Roche]) and sonicated (Bioruptor; Diagenode). The protein concentration was determined by Bradford assay, and 20 $\mu$g of protein were boiled in the presence of NuPAGE lithium dodecyl sulfate sample buffer (NP0007; Invitrogen) and NuPAGE Sample Reducing Agent (NU0004; Invitrogen) at 95°C for 5 min, resolved by SDS–PAGE (4–12% Criterion XT Bis-Tris Protein Gel; Bio-Rad), and transferred to nitrocellulose membrane (Bio-Rad). Staining with ATX Ponceau S Red (09189; Sigma-Aldrich) was used as a further marker of protein content. The membrane was then blocked with 5% non-fat milk in PBS-0.1% Tween-20 (P1379; Sigma-Aldrich) for 1 h at RT and probed with specific primary antibodies ($\alpha$-p21 [1:500, 556430; BD Biosciences], DIS3L [1:500, ab89042], and actin [1:1,000, A2103; Sigma-Aldrich]) overnight at 4°C. After three washes in PBS containing 0.1% Tween 20, the membrane was incubated with antirabbit or antimouse IgG HRP secondary antibodies for 1 h at RT and revealed by chemiluminescence, respectively. Detection was performed using Chemidoc MP imaging system (Bio-Rad). Experiments were performed in triplicate, and a representative Western blot was shown. Quantification of Western blot bands was performed using the ImageJ software.

## Mapping

For the mapping, SHRiMP was used (v2.2.3) (David et al, 2011) for the data set GSE55172 (color space reads) (parameters: -o 1 –max-alignments 10 –strata), whereas for the others, mapping was performed with STAR (v2.6.0b) (Dobin et al, 2013) (parameters: –outFilterMismatchNmax 1 –outSAMmultNmax 1 –outMultimapperOrder Random –outFilterMultimapNmax 30). Mapping was performed against the reference human genome (hg19 homo sapiens primary assembly from Ensembl) for the European Bioinformatics Institute data set, GSE55172, GSE81662, GSE85085, GSE108278, GSE130727, and against the reference mouse genome (mm9 mus musculus primary assembly from Ensembl) for the GSE77784 and GSE100535. The SAM files were then converted to the BAM format and sorted by coordinate with SAMtools (v1.7) (Li et al, 2009).

## Data observations

Bigwigs files were generated with bamCoverage (parameter: –normalizeUsing CPM) from deepTools (v3.1.3) (Ramirez et al, 2016). All observations were done using the Integrative Genomics Viewer software (Robinson et al, 2011).

## Differential gene expression

The package Subread (v1.28.1) (Liao et al, 2014) for R (v3.4.3) was used to count the uniquely mapped reads based on a gtf annotation file for hg19 or mm9 from Ensembl. Then the package DESeq2 (v1.18.1) (Love et al, 2014) was used to make the differential gene expression analysis and principal component analysis. $P$-values from the differential gene expression test were adjusted for multiple testing according to the Benjamini and Hochberg procedure. Only genes with an adjusted $P$-value lower than 0.05 were considered differentially expressed. GO term analysis were performed on these differentially expressed genes with Enrichr (Kuleshov et al, 2016).

## Heat maps and profiles

Reads inside upstream gene regions and downstream U snRNA regions were quantified using featureCounts (v1.6.1) from the Subread suite (Liao et al, 2014).

Then, from these counts, matrices were generated with the tool computeMatrix reference-point (parameter: –referencePoint TSS for the observations of the regions upstream of the genes or –referencePoint transcription end site for the observations of the downstream regions of U snRNA). Profiles were obtained using plotProfile (parameter: –perGroup) and heat maps using plotHeatmap (default paramaters) from the deepTools suite (Ramirez et al, 2016).

## Calculation of the Lo-Hi RNA stability index

The package Subread (v1.28.1) (Liao et al, 2014) for R (v3.4.3) was used to count the uniquely mapped reads based on a list of either highly unstable (t1/2 < 2 h) or highly stable (t1/2 > 10 h) mRNAs from Tani et al (2012). For each feature, the count was normalized using the Transcript per Million method. Then, for both conditions, the sum of every Transcript per Million was calculated and the highly unstable mRNAs was divided by the highly stable mRNAs, giving us a value for this Lo-Hi RNA stability index. Bed files of the highly unstable and highly stable mRNAs are available as supplemental data.

## Quantitative RT PCR and siRNA knock-down

Cells were transfected in six-well plates with Lipofectamine RNAiMax (Invitrogen, Thermo Fisher Scientific) and 10 nM siRNA duplexes (Sigma-Aldrich) following the manufacturer's instructions. Total RNA was prepared by guanidinium thiocyanate-phenol-chloroform extraction (Chomczynski & Sacchi, 2006). After DNAse treatment 20 min at 37°C, reverse transcription was carried out with SuperScript III (Invitrogen) and random hexanucleotides for 1 h at 42°C on 0.5 $\mu$g RNA, quantified with a nanodrop (Thermo Fisher Scientific). Real-time qPCR was carried out on a Stratagene Mx3005p with Brilliant III SYBR Green kits (Stratagene) according to the manufacturer's instructions.

## siRNAs

siEXOSC3 duplex 1: CACGCACAGUACUAGGUCA (Preker et al, 2008), siEXOSC3 duplex 2: GACGUCAGAUCAAAGAAAA, siNT: universal negative control #1 siRNA duplex (Sigma-Aldrich, ref SIC001).

## Primer sequences

CDKN1A-F CACTCAGAGGAGGCGCCATGTCA Amplicon Length: 213 bp
CDKN1A-R CCCAGGCGAAGTCACCCTCCA
CDKN2A-F CACCGAATAGTTACGGTCGG Amplicon Length: 129 bp

CDKN2A-R GCACGGGTCGGGTGAGAGTG
CDKN2B-F GGACTAGTGGAGAAGGTGCG Amplicon Length: 106 bp
CDKN2B-R GGGCGCTGCCCATCATCATG
OASL-F ATGTTGGACGAAGGCTTCACCACT Amplicon Length:193 bp
OASL-R ATCTGTACCCTTCTGCCACGTTGA
NLRP3-F CGATCAACAGGAGAGACCTTTAT Amplicon Length: 108 bp
NLRP3-R TGCTGTCTTCCTGGCATATC
DIS3L-F ACTCCGGGAATGTGCTAAAG Amplicon Length: 117 bp
DIS3L-R AGTAGCCTGTTCACAATGGG
EXOSC3-F CTCTCAGCAGAAGCGGTATG Amplicon Length: 103 bp
EXOSC3-R CTCACTCCCTCCAACATCAAC
GAPDH-F ATGGGGAAGGTGAAGGTCG Amplicon Length: 108 bp
GAPDH-R GGGGTCATTGATGGCAACAATA

# Data Availability

RNA-seq raw fastq files from GSE55172, GSE77784, GSE81662, GSE85085, GSE100535, GSE108278, and GSE130727 were downloaded from Gene Expression Omnibus on the National Center for Biotechnology Information database. The fastq files from E-MTAB-5403 were downloaded from ArrayExpress on the European Bioinformatics Institute database.

# Supplementary Information

# Acknowledgements

We thank Régis Courbeyrette for technical assistance and M Ricchetti for critical reading of the manuscript. C Muchardt was supported by grants from Institut Pasteur (PTR No 24-17), LABEX REVIVE, and Agence Nationale de la Recherche (ANR 15-CE14-0003). C Mann was supported by ANR 17-008-02.

## Author Contributions

N Mullani: investigation.
Y Porozhan: investigation.
A Mangelinck: investigation and methodology.
C Rachez: validation, investigation, methodology, and writing—review and editing.
M Costallat: formal analysis.
E Batché: investigation and methodology.
M Goodhardt: funding acquisition, methodology, and writing—review and editing.
G Cenci: funding acquisition, methodology, and writing—review and editing.
C Mann: conceptualization, formal analysis, funding acquisition, investigation, methodology, and writing—review and editing.
C Muchardt: conceptualization, data curation, formal analysis, supervision, funding acquisition, investigation, methodology, project administration, and writing—original draft, review, and editing.

## Conflict of Interest Statement

The authors declare that they have no conflict of interest.

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
