## [Reviewer comments · Life Science Alliance]

Life Science Alliance

Reduced RNA turnover as a driver of cellular senescence

Nowsheen Mullani, Yevheniia Porozhan, Adèle Mangelinck, Christophe Rachez, Mickael Costallat, Eric Batsché, Michele Goodhardt, Giovanni Cenci, Carl Mann, and Christian Muchardt

DOI: <https://doi.org/10.26508/lsa.202000809>

Corresponding author(s): *Christian Muchardt, Institut de Biologie Paris-Seine - FR3631*

Review Timeline:

Submission Date:	2020-06-08
Editorial Decision:	2020-07-28
Appeal Requested:	2020-08-13
Editorial Decision	2020-08-20
Revision Received:	2020-11-05
Editorial Decision:	2020-12-16
Revision Received:	2020-12-22
Accepted:	2020-12-23

Scientific Editor: Shachi Bhatt

Transaction Report:

July 28, 2020

Re: Life Science Alliance manuscript #LSA-2020-00809-T

Mr. Christian Muchardt
Institut Pasteur
Unité de de Régulation Epigénétique
25, rue du Dr Roux
Paris cedex 15 75724
France

Dear Dr. Muchardt,

Thank you for submitting your manuscript entitled "Reduced RNA turnover as a driver of cellular senescence". The manuscript has been evaluated by expert reviewers, whose reports are appended below. Unfortunately, after an assessment of the reviewer feedback, our editorial decision is against publication in Life Science Alliance.

As you will see, all referees acknowledge an overall interest in the proposed link between RNA degradation by the exosome complex and senescence. However, they are currently not convinced that the data analyzed so far provides sufficiently strong support for the conclusions. We will not reiterate all of the specific points here, but all referees are concerned about which conclusions can be drawn from the datasets, in part regarding the interpretation of effects (ref#1- point 1, 2) as well as the type (ref#2- point 2) and extent of the published sequencing data (ref#2- point 3). As referee #2 also notes, in our view, a dataset larger than two allowing for more statistical testing and/or additional experimental validation would be necessary to further substantiate the model, in particular as there appears to be variability for the observed phenotype depending on cell line or mode of senescence induction (ref #2- point 1; ref#3- point 5).

Although your manuscript is intriguing, we feel that the points raised by the reviewers are more substantial than can be addressed in a typical revision period. If you wish to expedite publication of the current data, it may be best to pursue publication at another journal.

Given the interest in the topic, we would be open to resubmission to Life Science Alliance of a significantly revised and extended manuscript that fully addresses the reviewers' concerns and is subject to further peer-review. If you would like to resubmit this work to Life Science Alliance, please contact the journal office to discuss an appeal of this decision or you may submit an appeal directly through our manuscript submission system. Please note that priority and novelty would be reassessed at resubmission.

Regardless of how you choose to proceed, we hope that the comments below will prove constructive as your work progresses. We would be happy to discuss the reviewer comments further once you've had a chance to consider the points raised in this letter.

Thank you for thinking of Life Science Alliance as an appropriate place to publish your work.

Sincerely,

Reilly Lorenz
Editorial Office Life Science Alliance
Meyerhofstr. 1
69117 Heidelberg, Germany
t +49 6221 8891 414
e contact@life-science-alliance.org
www.life-science-alliance.org

Reviewer #1 (Comments to the Authors (Required)):

The authors re-analyse published data reporting RNA sequencing in a range of senescence models and EXOSC3-depleted cells. The magnitude of the stabilization observed is not great, partially explaining why it was not previously much commented on in these datasets. The data and conclusions are somewhat disparate. However, the analyses appear to have been competently performed and the findings are of interest.

Specific comments:

1. Lo-Hi ratio. The results seem clear interpretation less so. If fast degradation is an active, energy dependent process whereas the slow degradation is the default, it could be that many insults will preferentially slow degradation resulting in the observed change in Lo-Hi ratio. This could be discussed.
2. Exosome subunit depletion, notable that the these are mRNA levels so changes in exosome abundance will lag behind. For transient treatment, significant changes in protein abundance would not be expected, unless turnover is also simulated by the insult.
3. P4 Results: The authors state: "RNAs transcribed downstream of genes are normally degraded by the RNA exosome, upon cleavage of the main transcript by the cleavage and polyadenylation apparatus (CPA)..." Previous analyses have concluded that the downstream fragments (which have free 5' ends) are degraded by Xrn2, as part of the "torpedo termination" mechanism. A reference and some discussion to support their claim would be helpful.
4. How do "upstream antisense RNAs (uaRNAs)" differ from the previously described PROMPTs, which are also upstream and antisense to protein coding genes? Is the inclusion of a further name warranted?

Minor points:

5. Page numbers would have been helpful.
6. References: Many need corrections: It looks like they were formatted as book sections.
7. P11: "patients with mutations in EXOSC2 are subject to premature ageing (32)...."
Reference missing.

Reviewer #3 (Comments to the Authors (Required)):

Mullani and colleagues, mostly using public datasets, show accumulation of RNA exosome substrates in senescent cells. This appears to be correlated with reduced RNA turnover. They show reduced expression of DIS3L (the catalytic subunit of the cytoplasmic RNA exosome complex) in senescent cells using those datasets. They also show upregulation of some components of antiviral machinery. Finally, they attempt to map SINE/LINE repeats and suggest that RNA transcribed from upstream of genes contains repeat seq. while the idea is interesting, the data are all correlative and no experimental validation is provided. Also, the biological relevance of the observation is not clear.

Data don't seem to be consistent between different cell types/senescence triggers. In particular, WI38 and IMR90 are very similar cells, and yet they fail to see the similar phenotype in IMR90 replicative/DNA damage senescence.

Fig. 1, RNA-seq: which dataset are these? in the text, they refer to the paper by Muniz, Deb et al., 2017, but according to the legend, the data seem to be from Lazorthes et al., 2015. The former is nascent RNA-seq, which is suitable for this fig. If the data are straight RNA-seq, the data are not strong enough and the interpretation would be difficult. Please clarify.

Fig. 2, RNA-seq: legend says "DESeq2 on RNA-seq data (n=2)". the data are not robust, and statistics should not be applied for n=2. The data would suggest some trend, then the authors need to experimentally validate it (e.g. by qPCR). And/or, there are numerous datasets for senescence in human fibroblasts (for this assay, straight RNA-seq should be sufficient), including the model they use here. Actually, they use a different cell system in Fig. 3a, again n=2, thus not very informative (without validation).

They show upregulation of some components of the antiviral OAS/RNASEL pathway. This appears consistent with the accumulation of unstable RNA. However, they state "The OASL gene, encoding a catalytically inactive isoform that retains double stranded RNA binding activity, displayed an even stronger activation." Such property implies that OASL might act as an endogenous dominant-negative factor. It would be important to functionally validate any increase in the activity of this pathway.

Fig. 5: "reduced RNA exosome activity has physiological consequences that resemble cellular senescence". This is not a typical senescence model and the data are merely correlative. This has to be experimentally validated. The authors could deplete those factors in human diploid fibroblasts and then conduct the standard senescence assays.

Reviewer #4 (Comments to the Authors (Required)):

In this manuscript the results indicate that in a subset of senescent cells, reduced RNA exosome activity has physiological consequences in driving and maintaining the permanent inflammatory state characterizing cellular senescence. In addition, the data are suggestive of a bidirectional crosstalk between RNA degradation and oxidative stress for the induction of growth arrest.

The conclusions are interesting but the manuscript requires some improvement. The manuscript under revision does not have page numbers nor line numbers what makes it difficult to indicate what needs modification.

Points that need to be revised:

In the whole text:

The work turnover usually is written as one word and not turn-over. Please correct in the introduction and throughout the text.

Make sure you give a good definition of uaRNAs and pegeRNAs to clarify the reader.

Abstract

Line 3_ "While cytoplasmic DNA was shown to drive the inflammatory phenotype of senescent cells, an equivalent role for RNA has never been explored" NEVER is not truth. The work of Myriam Gorospe lab and several others have connected RNA with senescence. Delete this sentence.

It would be important to stress even in the abstract your conclusion that reduced RNA turnover was not triggered by simple growth arrest (data from Fig3). And you have to indicate that it is only a subset of senescent cells that is affected.

Results and Discussion:

Under the sub-topic : "Reduced turn-over of unstable RNAs is observed in multiple senescent cells": In the Discussion you have tried to provide a possible explanation why only a subset of cells is affected. This is really puzzling and not clear to the rest of the story. See if you can improve it.

The role of long-noncoding RNAs in senescence is omitted from the Discussion. Please refer to the work done connecting RNA metabolism and types of RNA with senescence.

Thank you for the comments on our paper and thank you for giving us an opportunity to resubmit. I have now discussed the issue with the co-authors and it is our impression that most of the Reviewers comments can be addressed by rewriting and analysis of existing data, and that only one of the reviews calls for additional potentially time-consuming experiments. As a first step, we would like to put together an initial version of a rebuttal and also discuss with you the possibility of limiting additional experiments to one that would document the causal effect of RNA accumulation on the onset of senescence. Would you believe that this could be a sensible way to proceed?

With kind regards,

Christian Muchardt
Group Leader - Epigenetic Regulation
Dpt of Developmental and Stem Cell Biology
UMR3738 CNRS
Jacques Monod Bldg.
Institut Pasteur
25 rue du Dr-Roux
75724 PARIS CEDEX 15
FRANCE

MS: LSA-2020-00809-T

Mr. Christian Muchardt
Institut Pasteur
Unité de de Régulation Epigénétique
25, rue du Dr Roux
Paris cedex 15 75724
France

Dear Dr. Muchardt,

Thank you for submitting an appeal for your manuscript "Reduced RNA turnover as a driver of cellular senescence" [LSA-2020-00809-T]. I have read through your appeal note and re-evaluated the referees concerns.

I agree that most of what Rev 1 and 3 have pointed out can be addressed by reanalyzing some data and a rewrite. However, editorially, we agree with the concerns raised by Rev 2 - lack of validation and low sample size for any appropriate statistical analysis. The combined concerns from Rev 1 and 2 suggest that the conclusions are not robustly supported by the data. I am happy to reconsider a significantly revised manuscript, with a point-by-point rebuttal that clarifies how you have addressed the referees concerns, but please note that we can not provide any editorial guarantee without seeing the additional data. The revised manuscript will have to be reevaluated editorially, before deciding whether to send it back to the referees or not.

Yours sincerely,

Shachi Bhatt
Executive Editor
Life Science Alliance

Reviewer #1 (Comments to the Authors (Required)):

The authors re-analyse published data reporting RNA sequencing in a range of senescence models and EXOSC3-depleted cells. The magnitude of the stabilization observed is not great, partially explaining why it was not previously much commented on in these datasets. The data and conclusions are somewhat disparate. However, the analyses appear to have been competently performed and the findings are of interest.

Specific comments:

1. Lo-Hi ratio. The results seem clear interpretation less so. If fast degradation is an active, energy dependent process whereas the slow degradation is the default, it could be that many insults will preferentially slow degradation resulting in the observed change in Lo-Hi ratio. This could be discussed.

We agree that the causes of a modified Lo-Hi ratio can be numerous. We show the effect of EXOSC3 as an illustration of the impact of reduced RNA turnover on this ratio, without implying that a defective RNA exosome is the only possible cause of a modified Lo-Hi ratio. Thus, as indicated by the Referee, the Lo-Hi ratio is used as a simple way to evaluate an impact on active RNA degradation. In the new version, this is clearly mentioned in the discussion.

2. Exosome subunit depletion, notable that the these are mRNA levels so changes in exosome abundance will lag behind. For transient treatment, significant changes in protein abundance would not be expected, unless turnover is also simulated by the insult.

It is not clear to us which panel the Referee is referring to. All the experiments we described in the first version had a duration of at least 3 days, which should be sufficient for reduced mRNA production to have an impact on protein accumulation. In the case of the Wi38 cells driven into senescence by oncogenic RAF, Panel 2B confirms that the decreased levels in DIS3L mRNA correlates with decreased levels of the DIS3L protein.

3. P4 Results: The authors state: "RNAs transcribed downstream of genes are normally degraded by the RNA exosome, upon cleavage of the main transcript by the cleavage and polyadenylation apparatus (CPA)..." Previous analyses have concluded that the downstream fragments (which have free 5' ends) are degraded by Xrn2, as part of the "torpedo termination" mechanism. A reference and some discussion to support their claim would be helpful.

This part of the Result section has been rewritten and the statement has been removed. Instead, we are now bringing up the issue in the discussion, pointing out that reduced RNA exosome activity results in accumulation of reads downstream of genes, and we quote the studies by Lemay et al. 2014 and Villa et al. 2020, respectively showing direct and indirect implications of the RNA exosome in termination.

4. How do "upstream antisense RNAs (uaRNAs)" differ from the previously described PROMPTs, which are also upstream and antisense to protein coding genes? Is the inclusion of a further name warranted?

It is our understanding that uaRNAs and PROMPTs are two names for the same RNA species, respectively proposed by the Phil Sharp and the Alain Jacquier labs. In a new version, this is clearly stated. We coined the term pegeRNAs (perigenic RNAs) to describe transcripts

originating from both upstream and downstream of genes. PegeRNAs are thus composed of uaRNAs/PROMPTS plus downstream transcripts.

Minor points:

5. Page numbers would have been helpful.

We apologize for having forgotten to number the pages.

6. References: Many need corrections: It looks like they were formatted as book sections.

References have been checked.

7. P11: "patients with mutations in EXOSC2 are subject to premature ageing (32)..." Reference missing.

We apologize for this and we have now included this reference.

Reviewer #3 (Comments to the Authors (Required)):

Mullani and colleagues, mostly using public datasets, show accumulation of RNA exosome substrates in senescent cells. This appears to be correlated with reduced RNA turnover. They show reduced expression of DIS3L (the catalytic subunit of the cytoplasmic RNA exosome complex) in senescent cells using those datasets. They also show upregulation of some components of antiviral machinery. Finally, they attempt to map SINE/LINE repeats and suggest that RNA transcribed from upstream of genes contains repeat seq. while the idea is interesting, the data are all correlative and no experimental validation is provided. Also, the biological relevance of the observation is not clear.

Data don't seem to be consistent between different cell types/senescence triggers. In particular, WI38 and IMR90 are very similar cells, and yet they fail to see the similar phenotype in IMR90 replicative/DNA damage senescence.

Our exploration of various data sets does indeed suggest that the behavior of cells confronted with an inducer of senescence is very heterogeneous. This is consistent with the difficulty reported by several authors when it comes to define a transcriptional signature of senescent cells. Our data suggest the coexistence of mechanisms relying either on cytoplasmic DNA or double stranded RNA, or possibly on a combination of the two in a large variety of proportions. This could be a major source of heterogeneity in the transcriptional pattern from one cell line to the next and from one trigger to the other. Therefore, it seems reasonable to expect large variations between data sets, the constant being that either DNA detection or RNA detection or both get activated.

Fig. 1, RNA-seq: which dataset are these? in the text, they refer to the paper by Muniz, Deb et al., 2017, but according to the legend, the data seem to be from Lazorthes et al., 2015. The former is nascent RNA-seq, which is suitable for this fig. If the data are straight RNA-seq, the data are not strong enough and the interpretation would be difficult. Please clarify.

We apologize for this confusion in the references. The data analyzed in the figure are from Muniz, 2017, while Lazorthes, 2015 provides the initial description of the WI38 hTERT RAF1-ER cell line. This has been corrected. Concerning the use of RNA-seq vs nascent RNA-seq, we

respectfully disagree. Nascent RNA-seq allows following transcriptional activity as it monitors RNAs produced during a given period of time. In contrast, it does not provide information on RNA accumulation, which is what is needed here to examine RNA decay/stability.

Fig. 2, RNA-seq: legend says "DESeq2 on RNA-seq data (n=2)". the data are not robust, and statistics should not be applied for n=2. The data would suggest some trend, then the authors need to experimentally validate it (e.g. by qPCR). And/or, there are numerous datasets for senescence in human fibroblasts (for this assay, straight RNA-seq should be sufficient), including the model they use here. Actually, they use a different cell system in Fig. 3a, again n=2, thus not very informative (without validation).

The new Figures 2A to 2C now exclusively rely on data from RT-qPCR and Western blots. These data confirm the down-regulation of DIS3L and the upregulation of OASL and NLRP3.

Regarding the possibility of verifying the RNA-senescence connection in other datasets, this is essentially the purpose of Figure 3. As this Figure relies entirely on data mining, we cannot perform validation experiments. Yet, we hope that the Referee will agree that Fig3A is based on two different time points (2x2 biological replicates) showing a progressive decrease in expression of several RNA exosome subunits and a matching increase in OAS gene expression. We may also add that the p-values were calculated with DESEQ2 using the FDR/Benjamini-Hochberg approach. This p-value is usually considered as informative even when only two biological replicates are available, particularly because a low sample number has a strongly negative impact on this p value.

They show upregulation of some components of the antiviral OAS/RNASEL pathway. This appears consistent with the accumulation of unstable RNA. However, they state "The OASL gene, encoding a catalytically inactive isoform that retains double stranded RNA binding activity, displayed an even stronger activation. " Such property implies that OASL might act as an endogenous dominant-negative factor. It would be important to functionally validate any increase in the activity of this pathway.

As enunciated by the Referee, the purpose of monitoring OASL expression was the assessment of antiviral activity. We respectfully believe that exploring a possible dominant-negative activity of OASL is beyond the scope of the present study.

Fig. 5: "reduced RNA exosome activity has physiological consequences that resemble cellular senescence". This is not a typical senescence model and the data are merely correlative. This has to be experimentally validated. The authors could deplete those factors in human diploid fibroblasts and then conduct the standard senescence assays.

We have now depleted EXOSC3 with siRNAs in Wi38 cells harboring an inducible oncogenic Raf and we have then monitored expression of p15, p16, and p21 by RT-qPCR at different time points after induction of Raf. This experiment shows that reduced RNA exosome activity results in accelerated expression of the senescence markers p15 and p16 in the presence of Raf. In the new version, this experiment complements the initial panels showing that mouse embryonic fibroblasts inactivated for ExoSC3 display a transcriptional landscape that resembles that of senescent cells.

Reviewer #4 (Comments to the Authors (Required)):

In this manuscript the results indicate that in a subset of senescent cells, reduced RNA exosome activity has physiological consequences in driving and maintaining the permanent inflammatory state characterizing cellular senescence. In addition, the data are suggestive of a bidirectional crosstalk between RNA degradation and oxidative stress for the induction of growth arrest.

The conclusions are interesting but the manuscript requires some improvement. The manuscript under revision does not have page numbers nor line numbers what makes it difficult to indicate what needs modification.

We apologize for having forgotten the page numbers.

Points that need to be revised:

In the whole text:

The work turnover usually is written as one word and not turn-over. Please correct in the introduction and throughout the text.

This has been corrected.

Make sure you give a good definition of uaRNAs and pegeRNAs to clarify the reader.

These definitions have been clarified.

Abstract

Line 3_ "While cytoplasmic DNA was shown to drive the inflammatory phenotype of senescent cells, an equivalent role for RNA has never been explored" NEVER is not truth. The work of Myriam Gorospe lab and several others have connected RNA with senescence. Delete this sentence.

The abstract has been rewritten, eliminating this sentence. Furthermore, we are now quoting the work of the Gorospe lab in the introduction.

It would be important to stress even in the abstract your conclusion that reduced RNA turnover was not triggered by simple growth arrest (data from Fig3). And you have to indicate that it is only a subset of senescent cells that is affected.

The absence of an effect of growth arrest on RNA turnover is now mentioned at the end of the introduction (not in the abstract where it was difficult to put in the right context). We also now more clearly stress in the abstract, in the result section and in the discussion, that modified RNA turnover is seen only in a subset of senescent cells.

Results and Discussion:

Under the sub-topic : "Reduced turn-over of unstable RNAs is observed in multiple senescent cells": In the Discussion you have tried to provide a possible explanation why only a subset of cells is affected. This is really puzzling and not clear to the rest of the story. See if you can improve it.

This section has been rewritten and clarified.

The role of long-noncoding RNAs in senescence is omitted from the Discussion. Please refer to the work done connecting RNA metabolism and types of RNA with senescence. We now mention more extensively previous work on long non-coding RNAs.

December 16, 2020

RE: Life Science Alliance Manuscript #LSA-2020-00809-TR-A

Mr. Christian Muchardt
Institut Pasteur
Unité de de Régulation Epigénétique
25, rue du Dr Roux
Paris cedex 15 75724
France

Dear Dr. Muchardt,

Thank you for submitting your revised manuscript entitled "Reduced RNA turnover as a driver of cellular senescence". We would be happy to publish your paper in Life Science Alliance pending final revisions necessary to meet reviewer 1's concern about references and our formatting guidelines.

Along with the points listed below, please also attend to the following:

- please consult our Manuscript Preparation Guidelines <https://www.life-science-alliance.org/manuscript-prep> and put your manuscript sections in the correct order
- please add Author Contributions to the main manuscript text
- please add a separate conflict of interest statement to your main manuscript text
- please upload your supplementary figures as single files and add your supplementary figure legends to the main manuscript text directly under the main figure legends
- please add a callout for Supplementary Figure 3C,D in main manuscript text
- please add a legend for Panel G in Figure 3 in the figure legend
- we encourage you to revise the figure legends for figures 1, 2, 4, 5 such that the figure panels are introduced in an alphabetical order

A. FINAL FILES:

-- High-resolution figure, supplementary figure and video files uploaded as individual files: See our

detailed guidelines for preparing your production-ready images, <https://www.life-science-alliance.org/authors>

B. MANUSCRIPT ORGANIZATION AND FORMATTING:

Sincerely,

Shachi Bhatt, Ph.D.
Executive Editor
Life Science Alliance
<https://www.lsjournal.org/>
Tweet @SciBhatt @LSAJournal

Reviewer #1 (Comments to the Authors (Required)):

The MS has been only modestly revised and more could have been done to address the points raised. However, given the ongoing difficulties we are all experiencing in research, I am happy to support publication of the revised MS.

The references still need attention.

Reviewer #3 (Comments to the Authors (Required)):

the authors have addressed most of my questions.

December 23, 2020

RE: Life Science Alliance Manuscript #LSA-2020-00809-TRR

Dr. Christian Muchardt
Institut de Biologie Paris-Seine - FR3631
CNRS - UMR8256 - Biological Adaptation and Ageing
7-9, Quai Saint Bernard
Paris cedex 05 75252
France

Dear Dr. Muchardt,

Thank you for submitting your Research Article entitled "Reduced RNA turnover as a driver of cellular senescence". It is a pleasure to let you know that your manuscript is now accepted for publication in Life Science Alliance. Congratulations on this interesting work.

*****IMPORTANT:** If you will be unreachable at any time, please provide us with the email address of an alternate author. Failure to respond to routine queries may lead to unavoidable delays in publication.*******

DISTRIBUTION OF MATERIALS:

Again, congratulations on a very nice paper. I hope you found the review process to be constructive and are pleased with how the manuscript was handled editorially. We look forward to future exciting submissions from your lab.

Happy Holidays and a Happy New Year!!

Sincerely,

Shachi Bhatt, Ph.D.

Executive Editor

Life Science Alliance

<https://www.lsjournal.org/>
